# Methylglyoxal Impairs Sister Chromatid Separation in Lymphocytes

**DOI:** 10.3390/ijms23084139

**Published:** 2022-04-08

**Authors:** Leigh Donnellan, Clifford Young, Bradley S. Simpson, Varinderpal S. Dhillon, Maurizio Costabile, Peter Hoffmann, Michael Fenech, Permal Deo

**Affiliations:** 1Health and Biomedical Innovation, Clinical and Health Sciences, University of South Australia, Adelaide 5000, Australia; donly017@mymail.unisa.edu.au (L.D.); bradley.simpson@unisa.edu.au (B.S.S.); varinderpal.dhillon@unisa.edu.au (V.S.D.); maurizio.costabile@unisa.edu.au (M.C.); 2Clinical and Health Sciences, University of South Australia, Adelaide 5000, Australia; clifford.young@unisa.edu.au (C.Y.); peter.hoffmann@unisa.edu.au (P.H.); 3Centre for Cancer Biology, SA Pathology University of South Australia, Frome Road, Adelaide 5000, Australia; 4Genome Health Foundation, North Brighton 5048, Australia

**Keywords:** methylglyoxal, chromosomal instability, sister chromatid separation, proteomics

## Abstract

The accurate segregation of sister chromatids is complex, and errors that arise throughout this process can drive chromosomal instability and tumorigenesis. We recently showed that methylglyoxal (MGO), a glycolytic by-product, can cause chromosome missegregation events in lymphocytes. However, the underlying mechanisms of this were not explored. Therefore, in this study, we utilised shotgun proteomics to identify MGO-modified proteins, and label-free quantitation to measure changes in protein abundance following exposure to MGO. We identified numerous mitotic proteins that were modified by MGO, including those involved in the separation and cohesion of sister chromatids. Furthermore, the protein abundance of Securin, an inhibitor of sister chromatid separation, was increased following treatment with MGO. Cytological examination of chromosome spreads showed MGO prevented sister chromatid separation, which was associated with the formation of complex nuclear anomalies. Therefore, results from this study suggest MGO may drive chromosomal instability by preventing sister chromatid separation.

## 1. Introduction

The separation of sister chromatid into daughter cells is an incredibly complex process involving the coordinated efforts of hundreds of proteins [1]. Unsurprisingly, the errors that arise in this process can cause missegregation of chromosomes resulting in the generation of aneuploidy in the offspring [1,2]. As most tumours harbour aneuploid cells, mechanisms that lead to this may be causally implicated in tumour development [3]. Chromosome missegregation events can originate from a variety of factors including (i) erroneous or failed attachment of kinetochores to spindle microtubules [4], (ii) abnormal centrosome amplification and multipolar spindle formation [5], and (iii) defects in sister chromatid cohesion causing premature chromatid separation or inability of sister chromatids to separate [6,7].

We recently showed that methylglyoxal (MGO), a glycolytic by-product, is capable of inducing chromosome missegregation events in WIL2-NS B-lymphoblastoid cells and peripheral blood lymphocytes [8]. MGO is a highly reactive electrophilic metabolite that can react with the nucleophilic side chains of arginine and lysine residues on proteins, forming a variety of advanced glycation end-products (AGEs) such as N^ε^-(carboxyethyl)lysine (CEL) and N^δ^-(5-hydro-5-methyl-4-imidazolon2-yl)ornithine (MG-H1) [9]. Although it is well known that MGO can modify proteins, only several proteins have been identified as specific targets of MGO in robust biological and chemical detail. These include hypoxia-inducible factor-1α (HIF-1α), all canonical histone proteins, glyceraldehyde-phosphate dehydrogenase, and Kelch-like ECH-associated protein 1 [10,11,12,13]. Modification of proteins by MGO has previously been shown to alter protein function. For example, modification of histone proteins causes ablation of electrostatic interactions between histones and DNA due to the neutralisation of arginine and lysine positive charges, altered surface topology, and interference with other post-translational modifications [9,14,15]. Therefore, we hypothesised that MGO modification of proteins involved in the accurate segregation of sister chromatids may cause chromosome missegregation events.

To investigate this, we utilised liquid chromatography–mass spectrometry (LC–MS) to identify mitotic proteins harbouring MGO modifications. Furthermore, we performed label-free quantitation (LFQ) to determine the change in abundance of proteins after MGO exposure. Overall, we found numerous mitotic proteins to be modified by MGO, particularly those involved in the separation and cohesion of sister chromatids. Moreover, we observed that MGO increased the protein abundance of several cell cycle proteins involved in the separation of sister chromatids. Cytological examination of metaphase spreads confirmed the impairment of sister chromatid separation, which resulted in various, complex nuclear anomalies.

## 2. Results

### 2.1. Methylglyoxal Induced Micronuclei Formation Is Associated with Increased AGEs

In our previous study, we showed that MGO can induce the formation of micronuclei (MNi) and other DNA damage biomarkers (nuclear buds (Nbuds) and multipolar mitosis) in WIL2-NS cells after 48 h exposure [8]. As MGO can react rapidly with proteins, we investigated whether MNi formation was more pronounced at earlier time points and associated with the formation of MGO modifications. Therefore, WIL2-NS cells were exposed to 100 and 500 µmol/L MGO for 8, 24, and 48 h, followed by measurement of MGO modification levels (MG-H1 and CEL) and MNi. MG-H1 levels were significantly increased in cells exposed to 500 µmol/L MGO at all time points but not in cells exposed to 100 µmol/L (Figure 1A). CEL was significantly increased at both concentrations but only with 8 and 24 h treatments (Figure 1B). MNi were significantly increased at 8 and 24 h following exposure to both concentrations (Figure 1C). However, MNi were only elevated in cells exposed to 500 µmol/L at 48 h (Figure 1C). Correlation analysis was applied to explore the relationship between these MGO modifications and MNi formation. No significant correlation was observed for MG-H1; however, a significant correlation was observed for CEL (r = 0.693; *p* = 0.038) (Table 1). This suggests that CEL formation may be associated with the induction of MNi.

### 2.2. Methylglyoxal Modifies Various Components of the Mitotic Machinery

To better understand the role of MGO in MNi formation, which we previously observed was the result of missegregated chromosomes, a shotgun proteomics experiment was conducted to investigate both changes in protein abundance and the modification of mitotic proteins by MGO [8]. Based on time-point studies, cells were treated with 500 µmol/L of MGO for 24 h for all proteomics experiments. Although correlation analysis suggested CEL modifications were more closely associated with MNi formation, we also searched for MG-H (isomers 1, 2, and 3) and carboxethylarginine (CEA) modifications. A full list of proteins identified with MGO modifications is given in our previous study [16]. Enrichment analysis using DAVID was performed on the proteins identified with MGO modifications, which showed several overrepresented Reactome pathways, biological processes, and cellular components associated with mitosis (Figure 2A). Specifically, more than 50 cell-cycle-associated proteins were revealed to contain MGO modifications, as well as the involvement of sister chromatid cohesion, where the cohesin complex is one of the most enriched processes/pathways associated with these modifications.

We performed an LFQ study to ascertain changes in protein abundance resulting from MGO exposure, where 89 and 90 proteins exhibited a decrease or increase in abundance, respectively (Appendix A). Although enrichment analysis using DAVID on the downregulated proteins showed no downregulation of pathways associated with mitosis, several pathways associated with various mitotic processes were upregulated (Figure 2B). Similar to the results obtained from the enrichment analysis of MGO modifications, pathways associated with the regulation of sister chromatid cohesion and separation were increased in cells treated with MGO. An example includes PTTG1 (securin), which is an inhibitor of ESPL1 (separase), and a cysteine protease that cleaves the cohesin complex that adheres sister chromatids to each other (Figure 2C,D). Securin remains bound to separase until the onset of anaphase where it is ubiquitinated by the anaphase-promoting complex/cyclosome (APC/C): CDC20 complex targeting it for ubiquitin-mediated degradation, causing the release of separase and subsequent sister chromatid separation [17]. Other proteins, including CDC20, UBE2C, UBE2S, and BUB1, all of which are involved in the regulation of sister chromatid separation, were also upregulated (Figure 2C,D). The mitotic indices for untreated and MGO-treated cells were 7.67 ± 3.06% and 7.33 ± 1.53%, respectively, confirming the altered abundance of mitotic proteins was not due to the increased level of mitotic cells (Appendix A).

### 2.3. Methylglyoxal Modifies Various Components of the Mitotic Machinery

Proteomic analysis revealed the possibility that proteins involved in the separation of sister chromatids were susceptible to both modifications and increased protein abundance following MGO exposure and that this might cause disruption of the mitotic process. Therefore, using metaphase spreads, we investigated the resolution of sister chromatids, which is an indicator of their ability to dissociate from one another during the metaphase to anaphase transition. In control cells, we observed that the majority of metaphase spreads (51.1%) contained chromosomes in which sister chromatids were resolved from each other, whereas only 29.8% were ‘partially resolved’, and 19.1% were ‘unresolved’ (Figure 3A). In contrast, MGO exposure resulted in 61.1% of the metaphase spreads containing ‘unresolved’ chromatids, and only a small percentage (8.3%) were ‘resolved’. To further explore the basis of this observation, we investigated various nuclear anomalies (Figure 3), which have been shown to occur due to impairment of sister chromatid separation [6,18]. At 100 µmol/L, MGO increased the frequency of MNi, NPBs, fused, enlarged, and multinucleated cells after 24 h (Figure 3B). These biomarkers, along with cells containing multiple (≥2) MNi and Nbuds, were also increased at 500 µmol/L MGO (Figure 3B). Fused nuclei are well-described markers of failed sister chromatid separation and are believed to be caused by the presence of multiple ultrafine bridges which form when unseparated sister chromatids are pulled to opposite poles [19]. In this situation, centromeric signals should be present in, or very close to, the fusion region. Indeed, the fusion regions contained numerous centromeric signals, determined using fluorescence in situ hybridisation with pan centromeric probes (Figure 3C). This supports the hypothesis that fused nuclei are formed by the failed separation of sister chromatids and not by other mechanisms such as telomeric end fusion events which are less likely to contain centromeric signals.

## 3. Discussion

In this study, we found that MGO impairs sister chromatid separation in WIL-2NS cells, causing various nuclear anomalies associated with chromosomal instability (CIN). LC/MS revealed the identification of numerous mitotic proteins containing various MGO modifications, most of which were CEL modifications. Furthermore, we observed an increased abundance in several proteins involved in the regulation of sister chromatid separation. Most notably, we found treatment with MGO increased the protein abundance of the separase inhibitor, securin. Securin inhibits the cysteine protease separase until the onset of anaphase, at which point it is ubiquitinated by the anaphase-promoting complex/cyclosome (APC/C) ubiquitin ligase (E3) and degraded by the 26S proteasome system [20]. This process releases separase which cleaves the double-strand-break repair protein rad21 homolog (RAD21) of the cohesin complex which binds together sister chromatids, allowing them to separate. Overexpression or expression of non-degradable securin results in incomplete sister chromatid separation and gives rise to chromosomes (or sister chromatids) with an ‘unresolved’ phenotype (Figure 3A) [17,21]. Furthermore, increased abundance of securin has been shown to cause various nuclear anomalies, most notably MNi, NPBs, and multinucleated cells [17,21,22]. This is consistent with studies showing that expression of non-cleavable RAD21 (target of separase) prevents sister chromatid separation and causes similar morphologies as observed in this study [6]. Although prevention of sister chromatid separation by increased abundance of securin could account for the results observed in this study, other changes may also be contributing to the observed CIN. For example, cohesin can also be removed from chromosome arms by cleavage independent mechanisms [6]. During DNA replication in the S phase, soronin associates with chromatin, which is facilitated by acetylation of structural maintenance of chromosome 3 (Smc3) [23]. Soronin binds to Pds5, which alters the topology of Pds5 and Wap1, producing a stable closed cohesin complex [23]. Both sister chromatid cohesion protein PDS5 homolog B (Pds5B) and Smc3 were shown to contain MGO modifications [16]. While it is uncertain what impact this may have on cohesin function, altered topology due to MGO modification may cause structural changes in cohesin, resulting in an unresolved phenotype. Furthermore, overexpression of either CDC20 or UBE2C can increase the frequency of missegregated chromosomes, resulting in aneuploidy, polyploidy, and various other markers of CIN [24,25,26]. Overexpression of CDC20 was shown to promote premature anaphase, which is known to result in various aberrant phenotypes, including aneuploidy [24]. Moreover, increased abundance of securin, CDC20 and UBE2C were shown to be predictors of poor survival in patients with various cancers, including breast, colorectal, and gastric cancers [27,28,29,30,31].

We also identified numerous proteins with critical roles in mitosis to be modified by MGO. MGO has been shown to induce several changes in activity, modification of active/binding sites, and competition with other post-translational modifications [13,15]. MGO modifications have also been shown to cause structural changes in protein topology, likely caused by neutralising charged residues (arginine and lysine) which are involved in intramolecular interactions [15,32]. Therefore, modification of RAD21 or other cohesin complex proteins may have impaired cleavage by separase, preventing sister chromatid separation. Moreover, modification of other mitotic proteins, such as those involved in centrosome organisation or kinetochore–microtubule attachments, may have also contributed to the CIN observed in this study [4]. For example, impairment of the centrosome cycle can result in centrosome amplification (more than two centrosomes per cell) which causes multipolar spindle formation and merotelic (uneven) kinetochore–microtubule attachments [33]. Extra centrosomes can cluster together such that the cell undergoes bipolar mitosis; however, those chromosomes with merotelic kinetochore–microtubule attachments are prone to lagging at the metaphase plate to form MNi [33].

The presence of MGO modification alone cannot infer loss of protein function, and further studies are required to characterise which MGO modifications in mitotic proteins are strongly correlated with mitotic dysfunction and specific CIN phenotypes. Nevertheless, we showed that several critical proteins involved in mitosis are targets for MGO modification, particularly those involved in sister chromatid separation. Moreover, we showed MGO increased the abundance of securin, a separase inhibitor in which its increased abundance has been shown to cause CIN [21]. These results further unravel the MGO-induced proteomic changes that are likely to cause MNi and provide information on the underlying mechanisms of missegregated chromosome events observed after exposure to MGO. Many of the CIN events observed after exposure to MGO are the same as those associated with tumorigenesis [3,34]. Therefore, it is possible that MGO promotes tumorigenesis by driving increased CIN by preventing sister chromatid separation during mitosis.

## 4. Materials and Methods

### 4.1. Materials

All reagents, chemicals, and enzymes were purchased from Sigma (St. Louis, MO, USA) unless indicated otherwise. Isotopically labelled and unlabelled MG-H1, CEL, and lysine were purchased from Iris Biotech (Marktredwitz, Germany). Trypsin Gold (Promega, V5280) and ProAlanase (Promega, VA2161) were purchased from Promega (Madison, WI, USA).

### 4.2. WIL2-NS Cell Culture

The WIL2-NS (ATCC CRL-8155) cell line was kindly gifted by Commonwealth Scientific Research Organisation (Adelaide, Australia). WIL2-NS cells were cultured in complete RPMI-1640 medium supplemented with 5% (*v*/*v*) foetal calf serum (FCS), L-glutamine (1% *v*/*v*) and penicillin–streptomycin (1% *v*/*v*) at 37 °C in a humidified atmosphere with 5% CO_2_. Cells were seeded at 5 × 10^5^ cells/mL and incubated for 24 h before being treated with 100 and 500 µmol/L of MGO for 24 or 48 h (as indicated) for each experiment.

### 4.3. Identification of DNA Damage Biomarkers

Micronuclei (MNi) multiple MNi (≥2), nucleoplasmic bridges (NPBs), nuclear buds (NBUDs), and fused nuclei (Fused) were scored in binucleated cells using the cytokinesis-block micronucleus cytometry assay (CBMNcyt) [35]. Briefly, following treatment with MGO for 24 or 48 h, cells were washed twice with Hanks balanced salt solution and resuspended in RPMI-1640 containing 4.5 µg/mL cytochalasin-b (cyt-b) for 24 h. After cytokinesis block, cells were harvested onto slides by cytocentrifugation [35]. Slide preparation and scoring of the CBMNcyt assay were performed as described previously using a NanoZoomer S60 (Hamamatsu Photonics, Shizuoka, Japan) [35,36]. Enlarged, misshapen, and multinucleated cells were prepared the same as for the CBMNcyt assay, except cytochalasin-b was not added following treatment of MGO. Scoring criteria for the above biomarkers has previously been described [18,35]. For mitotic index, cells which possessed condensed chromosomes were distinguished from interphase cells microscopically, as previously described [37].

### 4.4. Quantification of MG-H1 and CEL in Whole-Cell Extracts

WIL2-NS cells were lysed in ice-cold RIPA buffer with sonication over ice for a 2 × 10 s burst (Misonix, NY, USA). Cellular debris was removed by centrifugation (17,000× *g* for 10 min). Protein was precipitated by the addition of ice-cold acetone (4:1 ratio of acetone: sample). The sample was left overnight at −20 °C before being centrifuged at 17,000× *g* for 10 min. Protein was washed twice with ice-cold acetone before resuspension in 50 μL of 50 mM ammonium bicarbonate (pH 8). Samples were incubated at 37 °C for 4 h to aid protein solubilisation. Undissolved protein was removed by centrifugation at 17,000× *g* for 10 min and protein concentration of the solution was measured by Bicinchoninic acid (BCA) assay following manufacturer’s instructions. Trypsin (TPCK treated sequencing grade) was added at an enzyme-to-protein ratio (1:50), and the sample was incubated at 37 °C for 16 h. Following incubation, the sample was heated to 95 °C for 10 min to denature the trypsin. Following this, the sample was cooled to room temperature, and pronase E and aminopeptidase were added at an enzyme-to-protein ratio (1:50) for a further 24 h. Enzymes were removed by the addition of ice-cold acetone, and the sample was centrifuged at 17,000× *g* for 10 min. The supernatant was collected, dried under vacuum centrifugation, and resuspended in 0.1% (*v*/*v*) formic acid containing an internal standard. The sample (2 μL) was injected, and analytes were separated using a 150 × 4.6 mm, 4 μm Phenomenex C18 column (Phenomenex, Torrance, CA, USA) with a linear gradient of 0.1% formic acid in water (Buffer A) and 0.1% formic acid in acetonitrile (Buffer B) over 5 min at a flow rate of 0.6 mL/min. Multiple reaction monitoring (MRM) was conducted in positive-ion mode using an AB Sciex 6500+ QqQ mass spectrometer with the following transitions *m*/*z* 147.4 > 83.9 (lysine), 151.2 > 87.9 (d4 lysine), 219.2 > 130.2 (CEL), 222.2 > 134.2 (d4 CEL), 229.2 > 116.1 (MG-H1) 232.2 > 116.1 (d3 MG-H1). The concentration of MG-H1 and CEL was normalised to lysine content and expressed relative to control.

### 4.5. Sister Chromatid Resolution Assay

Resolution of sister chromatids was scored in metaphase spreads of cells arrested in metaphase using Colcemid. Briefly, cells were treated with MGO for 24 h. Following treatment, colcemid (KaryoMAX, Thermofisher Scientific, Adelaide) was added to a concentration of 0.5 µg/mL for 2 h. After metaphase arrest, cells were gently centrifuged, and the supernatant was removed. Cells were resuspended in hypotonic 0.075 M KCl solution and incubated at 37 °C for 10 min. The supernatant was removed and ice-cold fixative (1:3 acetic acid: methanol) was added dropwise, and cells were incubated at 4 °C for 10 min. After the fixative was removed, cells were resuspended in approximately 100 μL of fresh fixative and dropped onto microscope slides. Slides were air-dried and stained as described previously [36]. Metaphase spreads were scored microscopically using an Olympus CX43 at a 100 × objective lens. Mitotic cells in metaphase spread experiments were categorised into three groups according to their morphology as previously described [38,39]: (1) resolved, where the arms of each sister chromatid were separated (i.e., resolved) but remained connected at the centromere; (2) partially resolved, where sister chromatids could be distinguished from one and other but were still connected along the length of the arms and centromere; (3) unresolved, where sister chromatids could not be distinguished from one and other and had an arm closed morphology. Metaphase spreads consisting of multiple morphologies were scored as the morphology accounting for the largest percentage of chromosomes.

### 4.6. Fluorescence In Situ Hybridisation Assay

Fluorescence in situ hybridisation (FISH), using FITC-conjugated peptide nucleic acid probes specific for pancentromeric regions of DNA, was used to investigate fused nuclei. Cells were prepared as described for CBMNcyt assay and FISH was performed, as previously described [40]. Slide images were acquired with a Zeiss CellDiscoverer 7 (Carl Zeiss AG, Jena, Germany) using a combination of two LED modules—385 and 495 nm wavelengths, in combination with a quad-band bandpass filter. A Plan-Apochromat 20×/0.8NA objective with a 1× tube lens and an Axiocam 506 monochrome camera were used for image acquisition. Acquired images were processed using ZEN Blue software (Carl Zeiss AG).

### 4.7. Proteomic Analysis of WIL2-NS

Samples were prepared for LC–MS as described previously, using trypsin and ProAlanase in separate digests [16]. Raw MS files were analysed using Proteome Discoverer 2.4 (Thermo Scientific). Data were processed with the Sequest HT search engine against a concatenated database containing the 74,811 forward entries from the UniProt human database (1 December 2019) and their respective decoy counterparts. Trypsin or ProAlanase (cleaves C-terminal to proline and alanine residues) was specified as the enzyme and a maximum of five missed cleavages was allowed. Precursor and fragment mass tolerances were 10 ppm and 0.02 Da, respectively. Arginine and lysine carboxyethylation (+72.021129), arginine methylglyoxal-hydroimidazolone (+54.010565), methionine oxidation (−15.994915), N-terminal acetylation (+42.010565), N-terminal methionine loss (−131.040485), and N-terminal methionine loss and acetylation (−89.02992) were set as dynamic modifications, with cysteine carbamidomethylation (+57.021464) designated a static modification. Protein and peptide false discovery rates were both set at 1%. LFQ was performed for trypsin and ProAlanase digests separately and a Log2 cut-off of 0.5 was applied. Enrichment analysis was performed using the Database for Annotation, Visualisation and Integrated Discovery (DAVID) v6.8 (https://david.ncifcrf.gov/) accessed 18 January 2022 [41].

### 4.8. Data Analysis

Data are expressed as mean ± SD of three independent experiments (*n* = 3). A one-way or two-way analysis of variance (ANOVA), followed by a Dunnett post hoc test, was conducted (depending on experimental design) to determine statistical significance using GraphPad Prism (San Diego, CA, USA, Version 8.3.0). A *p* < 0.05 was considered significant. Two-tailed Pearson correlations were used to analyse relationships between AGE biomarkers and MNi frequency.

## Figures and Tables

**Figure 1 ijms-23-04139-f001:**
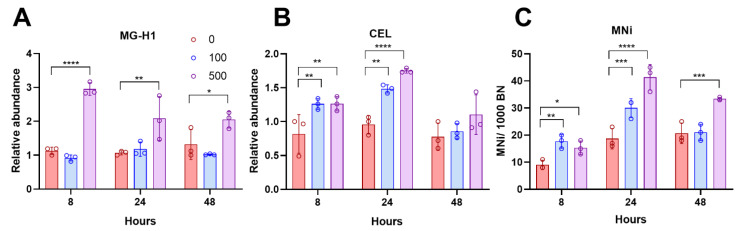
(**A**) MG-H1 concentration in whole-cell extracts at 8, 24, and 48 h normalised to control; (**B**) CEL concentration in whole-cell extracts at 8, 24, and 48 h normalised to control; (**C**) MNi formation, measured by CBMNcyt assay at 8, 24, and 48 h expressed as number of MNi/1000 Binucleated (BN) cells. Data are mean ± SD (*n* = 3). A two-way ANOVA followed by a Dunnett post hoc was conducted to determine statistical significance. * *p* < 0.05, ** *p* < 0.01, *** *p* < 0.001, **** *p* < 0.0001.

**Figure 2 ijms-23-04139-f002:**
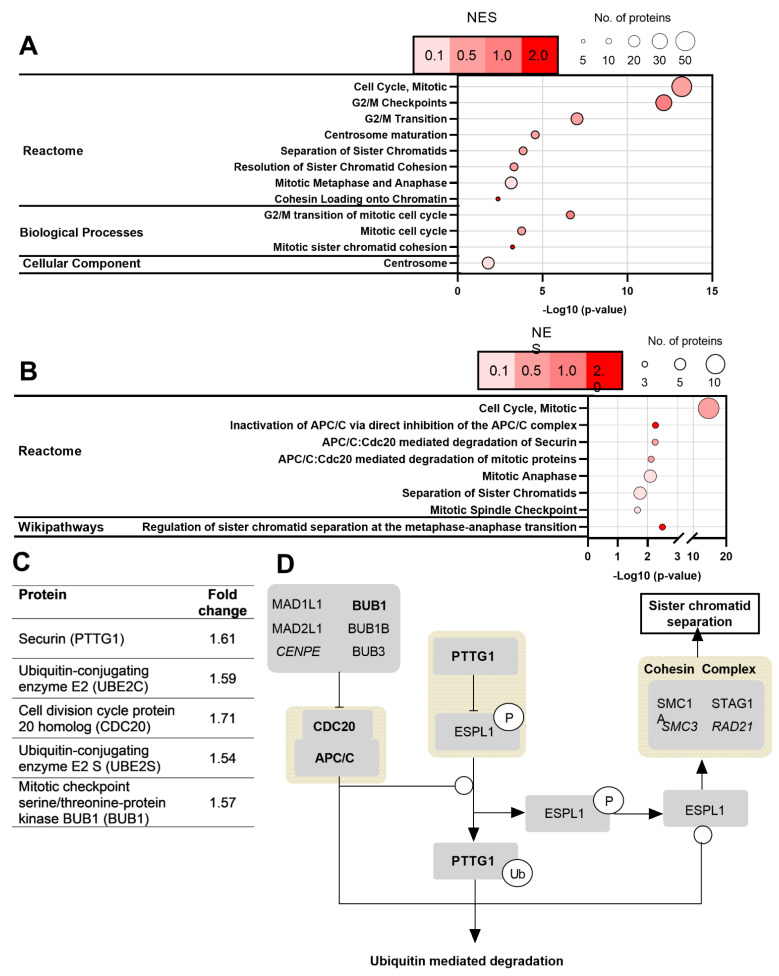
(**A**) Enrichment analysis of MGO-modified proteins using DAVID. Due to a large number of enriched terms, only those associated with mitosis were included and redundant terms were excluded. NES, normalised enrichment score. NES was calculated by dividing the fold enrichment by average enrichment of all terms; (**B**) enrichment analysis of upregulated proteins following MGO treatment (500 µmol/L for 24 h). Due to a large number of enriched terms, only those associated with mitosis were included and redundant terms were excluded; (**C**) fold change in proteins of interest following MGO treatment; (**D**) diagram showing the regulation of sister chromatid separation. Proteins in bold show increased abundance by MGO. Proteins in italics contain one or more MGO modifications.

**Figure 3 ijms-23-04139-f003:**
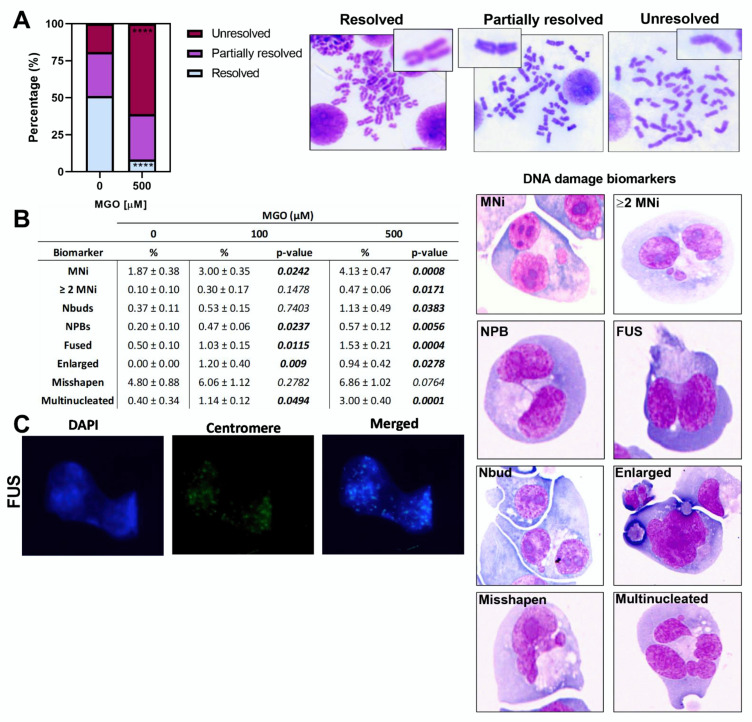
(**A**) Analysis of sister chromatid resolution, scored as (i) resolved, (ii) partially resolved, and (iii) unresolved; (**B**) analysis of DNA damage biomarkers. Columns show the percentage of cells displaying each biomarker and its significance determined by one-way ANOVA when compared with the control. Representative photomicrographs of each nuclear anomaly are displayed on the right. MNi, ≥2 MNi, NPB, FUS, and Nbuds were scored in binucleated cells using the CBMNcyt assay. Enlarged, misshapen, and multinucleated cells were scored in normally dividing cells; (**C**) fluorescence in situ hybridisation analysis of fused nuclei. Data are mean ± SD (*n* = 3). A one-way ANOVA, followed by a Dunnett post hoc was conducted to determine statistical significance **** *p* < 0.0001.

**Table 1 ijms-23-04139-t001:** Pearson correlations for relationships between AGE biomarkers and MNi.

		**MG-H1**	**CEL**
**MNi**	r	0.26	0.693
	*p*	0.495	0.038

Two-tailed Pearson correlations were used to analyse relationships between AGE biomarkers and MNi frequency.

## Data Availability

Raw MS files have been deposited to the ProteomeXchange Consortium via the PRIDE partner repository with the dataset identifier PXD032812 [42].

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
