# Peer review of "Methylglyoxal Impairs Sister Chromatid Separation in Lymphocytes"

_ijms, 2022, doi:10.3390/ijms23084139_

Round 1
Reviewer 1 Report
In their previous study, they proposed that methylglyoxal (MGO) caused chromosome missegregation events in lymphocytes, then Donnellan et al. aim to reveal this mechanism utilizing the proteomics approach in this study. They found that MGO specific modifications on many mitotic-related proteins and the proteins associated with various mitotic processes were increased in the cells following treatment with MGO. They also found the defects of sister chromatid separation in the presence of MGO. Therefore, they propose that MGO dependent change of protein abundance and modifications on the mitotic related proteins promotes tumorigenesis by preventing proper sister chromatid separation during mitosis.
The proteomic data are valuable to reveal which proteins are modified and altered their function following MGO exposure to understand the MGO induced nuclear defects. However, several points are still unclear and should be addressed before publication.
Major points
The authors performed enrichment analysis for the MGO modified proteins and the upregulated proteins following MGO exposure. They found that many proteins associated with mitosis were increased in both analyses. However, they do not describe the control detail for the data normalization in those analyses. It is possible to increase the counts of mitotic proteins due to the high mitotic cell population following MGO treatment. Does the mitotic index increase following MGO exposure? If so, how the MS counts are normalized and are evaluated correctly?
In Fig3A, the authors performed chromosome spread analysis to examine the sister chromatid cohesion in cells arrested at prometaphase with colcemid. They set the criteria of the cohesion status, “Resolved,” “Partially resolved,” “Unresolved” at prometaphase and show that the MGO exposure affects the arm resolution. They describe the possible relationship between upregulation of the securin amount and the arm resolution phenotype. The simple interpretation is that the phenotype is not caused by separase-dependent cohesin degradation since the spindle checkpoint is an “on” condition because of the presence of colcemid. In this condition, the securin is also stable in the absence of MGO. Instead, the arm resolution phenotypes may depend on the cohesin release from the chromosome arm via Plk1 and Cdk1 dependent phosphorylation of cohesin and sororin (prophase pathway). Another possibility is that incomplete DNA replication at S-phase resulted in unresolved sister chromatids at mitosis. They should clarify the above points.
In Fig3C, the authors performed FISH with a centromere marker in the fused nuclei to see the position of the centromere signal. They propose that centromere signals at the fusion region suggest the defect of the separation of sister chromatid. However, the cell cycle stage of the presented nuclei is unclear, and it is difficult to conclude the type of chromosome missegregation based on FISH in Fig3C because chromosomes or centromeres might move to another place in the nucleus after entering into the next G1. Therefore, it is better to examine the chromosomes at the metaphase to anaphase without the spindle poison with the appropriate centromere marker, such as immuno-staining with a centromere protein antibody. The behavior of the chromosomes at meta/ana transition provides valuable data to confirm the sister chromatid separation defect and distinguish the reasons these defects occur.
Minor points
Line 65, 66:
The authors use the abbreviation “MNi” without a description.
MNi means micronuclei?
Reference 8, 14 may be incomplete information.
Reviewer 2 Report
Comments are in the attached file.
They are technical ones, can be easily corrected, and after that the paper can be published in the Int. J. Mol. Sci. journal.

Reviewer 3 Report
The current manuscript is a well planned research output. The authors have provided enough experimental evidences to prove their hypothesis. The manuscript is excellent and I would recommend to publish in IJMS.
I would suggest to provide representative images (for all the experiments where possible) for MGO (0, 100, 500 uM etc.) for better illustration of the study results.
Line 308, is that necessary here (manuscript under review)? If under review, I think we can't write has been previously described. The current manuscript could be published before that one. In this manuscript, please provide details as much required to conduct the study independently in materials and methods.
Round 2
Reviewer 1 Report
The revised manuscript has been improved and addressed all my concerns.